# Pneumococcal Pneumonia and Invasive Pneumococcal Disease in Those 65 and Older: Rates of Detection, Risk Factors, Vaccine Effectiveness, Hospitalisation and Mortality

**DOI:** 10.3390/geriatrics6010013

**Published:** 2021-02-04

**Authors:** Roger E. Thomas

**Affiliations:** Department of Family Medicine, Cumming School of Medicine, University of Calgary, Calgary, AB T2N 4N1, Canada; rthomas@ucalgary.ca; Tel.: +1-403-607-1604

**Keywords:** pneumococcal pneumonia, invasive pneumococcal disease, risk factors for PP and IPD, seniors, vaccination effectiveness, pneumococcal nasal and pharyngeal carriages rates, PCV and PPV23 pneumococcal vaccine effectiveness, detection methods for PP and IPD, mortality, nursing home staff vaccination rates, nursing home redesign to minimise transmission

## Abstract

Pneumococcal pneumonia (PP) and invasive pneumococcal disease (IPD) are important causes of morbidity and mortality in seniors worldwide. Incidence rates and serious outcomes worsen with increasing frailty, numbers of risk factors and decreasing immune competence with increasing age. Literature reviews in Medline and Embase were performed for pneumococcal disease incidence, risk factors, vaccination rates and effectiveness in the elderly. The introduction of protein-conjugated pneumoccal vaccines (PCV) for children markedly reduced IPD and PP in seniors, but serotypes not included in vaccines and with previously low levels increased. Pneumococcal polysaccharide (PPV23) vaccination does not change nasal and pharyngeal carriage rates. Pneumococcal and influenza vaccination rates in seniors are below guideline levels, especially in older seniors and nursing home staff. Pneumococcal and influenza carriage and vaccination rates of family members, nursing home health care workers and other contacts are unknown. National vaccination programmes are effective in increasing vaccination rates. Detection of IPD and PP initially depend on clinical symptoms and new chest X ray infiltrates and then varies according to the population and laboratory tests used. To understand how seniors and especially older seniors acquire PP and IPD data are needed on pneumococcal disease and carriage rates in family members, carers and contacts. Nursing homes need reconfiguring into small units with air ventilation externally from all rooms to minimise respiratory disease transmission and dedicated staff for each unit to minimise transmision of infectious diseaases.

## 1. Introduction

### 1.1. The Burden of Pneumococcal Disease and Issues in Reducing Incidence Among Seniors

Pneumococcal pneumonia (PP) and invasive pneumococcal disease (IPD) are important causes of morbidity and mortality in seniors worldwide. In the nasopharynx the pneumococci secrete biofilms and the pneumococcal bacterial capsule facilitates their survival in colonised individuals by avoiding host immunological mechanisms that recognise pneumococcal capsular proteins. However, when pneumococci penetrate the host’s epithelium the colonies change from rough colonies to smooth colonies as the pneumococci upregulate their expression of pneumococcal capsular proteins. The bacteria then migrate either to mucosal surfaces in the ear causing otitis media or the lungs causing pneumonia or are transmitted by blood vessels to multiple sterile sites causing invasive pneumococcal disease (IPD) in joints, bones, the heart, the peritoneal cavity, brain or lungs. However, most pneumococcal pneumonias are not associated with detected bacteremia [1]. 

There are 98 serotypes in 21 serogroups with each serogroup containing 2–8 serotypes with similar capsules. The capsule protects the pneumococcus from host cell phagocytosis and is thus the key factor in virulence. Multiple molecules in the cell wall, cell membrane and cytoplasm also contribute to pneumococcal pathogenesis and thus virulence. Pneumococci secrete a cytotoxin Pneumolysin which increases pathogenesis and *lytA* expresses Autolysin and the *ply* gene expresses Pneumolysin. Cell wall proteins also interfere with the activity of the complement pathway, reduce deposition of complement and prevent opsonophagocytosis and lysis. Pneumococcal H inhibitor (Hic) impedes C3 convertase formation, PspA and CbpA inhibit C3b deposition, and pneumococcal surface protein C (PspC) binds factor H and probably accelerates C3 breakdown. Sialidase NanA cleaves sialic acid on host cells and adhesion to host cells is facilitated by multiple pneumococcal proteins including pneumococcal surface adhesin A (PsaA) [2]. 

Nasopharyngeal pneumococcal carriage rates in children <5 years of age range from 20% to 50% (varying principally with the intensity of exposure to other children), and in young and in middle-aged adults range from 5% to 15% but intermittently there are higher rates [3]. Pneumococcal pneumonia is often a result of co-infections. A review of the pathological processes in progression to pneumococcal pneumonia concluded:

“evidence continues to accrue that pneumococcal pneumonia events are often the result of co-infection with viral or other bacterial pathogens. Thus a pneumonia case resulting from a pulmonary infection with a single pathogen is probably an uncommon event; rather, most cases of pneumonia likely result from the sequential or contemporaneous co-infection of a host with multiple pathogens, often both viruses and bacteria.” [4]. 

“Intercurrent viral infections make the host more susceptible to pneumococcal colonization, and pneumococcal disease in a colonised individual often follows perturbation of the nasopharyngeal mucosa by such infections. Local cytokine production after a viral infection is thought to upregulate adhesion factors in the respiratory epithelium, allowing pneumococci to adhere via a variety of surface adhesin molecules, including PsaA, PspA, CbpA, PspC, Hyl, Pneumolysin, and the neuraminidases.” [5]. 

Multiple risk factors predispose the host to pneumococcal illness: asplenia or splenic dysfunction; chronic respiratory disease; chronic heart disease; chronic kidney disease (including renal transplantation); chronic liver disease; diabetes mellitus managed with insulin or medications; immunocompromise (including HIV, myeloma, generalised malignancy, chemotherapy, organ transplantation, bone marrow transplantation, and systemic glucocorticoid treatment for >1 month at a dose equivalent to ≥20 mg/d [children, ≥1 mg/kg per day]) [6]. 

There are seven important issues in assessing the effectiveness of pneumococcal vaccination and the role that it should play in the elderly. 1. what is the relative effectiveness of individual vaccinations and sequences of pneumococcal polysaccharide (PPV) and pneumococcal conjugated (PCV) vaccines in the elderly? 2. To what degree do pneumococcal vaccines generate enough antibody response and protect seniors at high risk of respiratory illnesses during respiratory disease seasons, including seniors with multiple risk factors and those in nursing homes and long-term care facilities? 3. What is the evidence for the effectiveness of booster vaccinations? 4. Mortality tables show that mortality rates increase progressively in the age groups 65–69, 70–74, 75–79, 80–84, 85–90 and >90 and data are needed about pneumococcal vaccine effectiveness for these age groups. 5. The introduction of pneumococcal vaccines for children has had profound effects in changing the pneumococcal environment for seniors. Continued monitoring of this relationship is needed as the vaccines change the prevalence of both serotypes currently reduced by vaccines and serotypes not currently treated by vaccines whose prevalence may increase. 6. Can the role of vaccination be enhanced for seniors and those who come into contact with them during respiratory disease seasons by increasing rates of consistent social distancing and use of protective personal equipment (gloves, gowns, face masks and face shields)? [7]. 7. Can the role of vaccination be supplemented by a major focus on building new nursing homes with multiple separate small units with dedicated staff for each unit so that transmission of respiratory infections is minimised by reducing the numbers of other patients, staff members, family members or visitors who contact each patient? Can current nursing homes be redesigned to decrease transmission from visitors, staff and other patients? They are often multi-story with the entire senior patient population visited by the same nursing care, personal care, pastoral care/social work, food, laundry, and maintenance staff. This would involve remaking them into small self-contained units, redesigning air transmission systems and changing staffing patterns to minimise individual staff members caring for multiple patients.

### 1.2. Purposes

To assess for the age groups 65–74, 75–84, 85–90 and >90 who have increasing frailty and decreasing immunocompetence the incidence of pneumococcal pneumonia (PP) and invasive pneumococcal disease (IPD), detection methods, carriage rates, risk factors, vaccination effectiveness, and rates of hospitalisation and death stratified by vaccination status and risk factors

## 2. Materials and Methods

### Literature Searches

Literature searches were performed on 1 December and updated on 25 December 2020 in Medline using the terms (pneumococcal vaccines) and (randomized controlled trial or cohort study or systematic review) [1841 citations] and in Embase using the terms (pneumococcal vaccines) and (randomized controlled trial or cohort analysis or systematic review) [2106 citations]. Additional searches were performed on 27 December on (pneumococcal vaccines or pneumococcal vaccination) and (vaccination coverage or vaccination rates). 

## 3. Results

### 3.1. Literature Review

Three systematic reviews of pneumococcal vaccines were identified by Remschmidt (2016) [8], Falkenhorst (2017), [9] and Berild (2020) [10]. Remschmidt identified two randomised controlled trials (RCTs) (but only one arm had relevant data so those arms were treated as cohort studies), eight prospective and three retrospective cohort studies and one cross-sectional design. The authors concluded that 11 (78%) studies were at high risk of bias due to selection bias and insufficient control for confounders and three from using non-specific ICD-9 disease codes within passive surveillance systems and possible selection bias during recruitment [8].

Falkenhorst’s systematic review identified four RCTs (four were rated with the Cochrane risk of bias tool as low risk of bias for IPD, but only two for the outcome of PP), and of thirteen observational studies (five cohort studies, three case control studies and five case studies) ten were judged at low risk of bias. For IPD for the RCTs the pooled vaccine effectiveness (VE) was 73% (95% confidence interval (CI) 10%, 90%; I^2^ = 11%) and for the cohort studies at low risk of bias 45% (15%, 65%, I^2^ = 0%). For PP for the two RCTs rated as low risk for this outcome VE was 64% (35%, 90%; I^2^ = 0%) and for the cohort studies 48% (25%, 63%, I^2^ = 0%) [9]. The GRADE [11] quality of evidence was moderate for the RCTs, with downgrading due to imprecision (wide confidence intervals) for IPD and for indirectness because the evidence was mainly based on one study of very old and frail nursing home residents in whom the VE would be different from that of the general population ≥60 [3]. Only one study [12] was reported in both Remschmidt’s and Falkenhorst’s systematic reviews.

Berild’s search was limited to the period January 2016 to April 2019 after these two systematic reviews and reported five articles which reported The Netherlands CAPiTA RCT, and nine observational studies (three cohort studies, three test-negative designs and three case control studies) of which four were judged on the Newcastle-Ottawa scale as at high risk of bias. Studies were reported individually and no meta-analyses were performed [10]. For data sources please see Appendix A. Data sources.

### 3.2. Prevalence of Pneumococcal Disease

In the US in 2019 there were an estimated 502,600 nonbacteremic pneumococcal pneumonia cases, 29,500 IPD cases and 25,400 pneumococcal-related deaths among the 91.5 million adults >50 years [13]. The prevalence of pneumococcal disease is substantially greater in seniors than in younger individuals.

Rates of pneumococcal disease for the UK between 1990 and 2015 were assessed by a systematic review of 38 prospective, retrospective, registry and surveillance cohorts based on patients admitted to hospital (no studies of outpatients were identified). In 2013/14 the rate of invasive pneumococcal invasive disease (IPD) was 6.85/100,000 for all adult age groups and 20.58/100,000 for those >65, and for community acquired pneumonia (CAP) and non-invasive disease was 20.6/100,000 adults. There were approximately 192,281 hospital admissions for pneumonia and 6000 cases of IPD in the UK in 2013/14 [14].

The incidence of pneumococcal disease in seniors is increased by several risk factors. A survey in UK general practice in 2009 estimated that in patients ≥65 with no risk factors the incidence of CAP was 17.9/100,000, for the 44.8% of patients ≥65 who had at least one risk factor 48/100,000, for the risk factor of Chronic Obstructive Pulmonary Disease (COPD) 91/100,000 and for chronic liver disease 129/100,000. Several serotypes had high fatality rates for those ≥65: 3 (39%), 31 (40%), and 19F (41%) [15].

In the UK the pneumococcal polysaccharide vaccine (PPV23) was authorised for patients ≥80 in 2003, ≥75 in 2004/5 and ≥65 in 2005/6. For children the first pneumococcal conjugate vaccine PCV7 was introduced in 2007 and PCV13 in 2010. After the introduction of the PCV vaccines for children there was a beneficial herd protection effect on rates of pneumococcal diseases in seniors. For those >65 years in 2008–2010 the incidence of PCV13 IPD serotypes was 10.33/100,000 and declined to 3.72/100,000 in 2013/14 [16].

PCV13 was introduced for children in most Canadian provinces in 2010 and the Canadian Immunization Research Network (CIRN) Serious Outcomes Surveillance (SOS) Network conducted active surveillance of CAP and IPD in hospitalised adults from December 2010 in hospitals in five provinces. This identified both reductions in pneumococcal disease rates for seniors and also which serotypes were most responsible. The study identified 6687 CAP cases with 15.9% *S. pneumoniae* positivity reported in 2011, 8.8% in 2014 and 12.9% in 2015, with PCV13 serotypes involved in 8.3%, 4.6% and 6.3% of cases respectively. Declines were attributed mostly to declines in serotypes 7F and 19A with a proportional increase in serotype 3. Of the admitted patients with a positive blood or sputum culture 54% of the cohort admitted in 2010–2013 had previously received a pneumococcal vaccine and 57% of the 2014–2015 cohort and 30-day mortality was 9.7% in 2010–2013 and 5.9% in 2014–2015. The study demonstrates the benefits of herd protection via child vaccination and the importance of specific serotypes [17].

PCV7 was introduced into the Belgian national vaccination programme in 2007 and national laboratory records of pediatric IPD incidence, serotypes and antimicrobial susceptibility from 2007–2018 were examined. Belgium had a unique PCV pediatric vaccination sequence beginning with PCV7 in 2006 (42% uptake), PCV7 in 2009–10, PCV13 in 2013–14, both PCV13 and PCV10 in 2015–16 and PCV10 was resumed in 2017–18 on expert advice of equal benefit from PCV10. The proportion of PCV7 serotype cases decreased from 17% in 2007 to 2% in 2009 and 3% in 2010. During the PCV7 period the average incidence of IPD was 22.8/100,000 children younger than 16 years, during the PCV13 period 11.7 cases/100,000, and during the PCV13-PCV10 period 9.2/100,000. Then in the 2017–19 PCV10 period the rate rose to 11.2 cases/100,000 children, mainly due to an increase in serotype 19A cases, and the dominant 19A clones differed from those in previous PCV periods. The authors commented “our observation underlines the importance of constantly following up on the dynamics of pneumococci to guide decision making in vaccine policies” [18].

### 3.3. Pneumococcal Vaccination Rates

There is no single document which provides world-wide pneumococcal vaccination rates through 2019 for those ≥65. Pneumococcal vaccination rates in those ≥65 are suboptimal. In 2015 in the US the vaccination rate was 60.2% for those 65–74, 68.6% for 75–84 and 68.3% for 85+; 41.7% for Hispanics, 68.1% for non-Hispanic whites, 50.2 for non-Hispanic Blacks, 49% for non-Hispanic Asians; and 48.7% for poor and 66% for non-poor individuals (CDC poor and non-poor classification) [19].

A retrospective cohort of >26 million US Medicare fee-for-service patients ≥65 2015–2017 is one of the few studies that provides a detailed listing by quintiles of vaccination rates for those ≥65 to 100+: Pneumococcal vaccination rates were: 65–69 (47.5%), 70–74 (49.7%), 75–79 (49.7%), 80–84 (48.1%), 85–89 (45.5%), 90–94 (39.7%), 95–99 (32.5%), and 100+ (15.1%); Influenza vaccination rates were higher at more advanced ages than the pneumococcal rates: 65–69 (44.2%), 70–74 (52.2%), 75–79 (56.3%), 80–84 (57.9%), 85–89 (57.9%), 90–94 (54.8%), 95–99 (49.9%), and 100+ (35.8%) [20].

For US nursing homes the Minimum Data Set of the US Centers for Medicare and Medicaid Services found pneumococcal vaccination coverage increased from 67.4% in 2006 to 78.4% in 2014 and influenza vaccination coverage increased from 71.4% in the 2005–2006 influenza season to 75.7% in the 2014–2015 season but there were large variations in pneumococcal vaccination coverage by state in 2014 (55.0% to 89.7%) and influenza vaccination coverage (50.0% to 89.7%) in the 2014–2015 influenza season [21].

A retrospective population-based observational study in January 2017 of 2,057,656 individuals ≥50 years olds in primary care centres in Catalonia, Spain, found that 796,879 (38.7%) had received PPV23 including 9.2% (95,409/1,039,872) of 50–64 year olds, 63.1% (434,408/688,786) of 65–79 year olds and 81.2% (267,062/328,998) of ≥80 year olds (*p* < 0.001). However, only 13,607 (0.7%) had received PCV13 [22].

In a sample of 2,531,227 individuals ≥15 years in the Shanghai Centers for Disease Control and Prevention information systems on chronic disease management, hospital records, and immunizations 22.8% were vaccinated for pneumonia from January 2013 to July 2017 but only 0.4% for influenza during the 2016/17 influenza season [23].

A study of nearly 10,000 IPD cases in those 65 and older in England and Wales 2012–2016 found that PPV23 vaccination effectiveness was 27% (17, 35) after adjusting for age, comorbidities and infection year. Vaccine effectiveness did not vary significantly with the interval after vaccination, and was 41% (23, 54) for those vaccinated within two years, 34% (15, 48) for those vaccinated 2–4 years previously, and 23% (12, 32) for those vaccinated ≥5 years previously. Vaccine effectiveness was 45% (27, 59) in those with no risk factors, 25% (11, 37) in high-risk immunocompetent patients and 13% (−9, 30) in the immunocompromised patients (difference *p* = 0.05) [24].

National vaccination programmes have been an important stimulus in increasing pneumococcal vaccination rates. In Japan pneumonia is the third most common cause of death but pneumococcal vaccination rates were low until the government started a public subsidy programme in 2014 for PPV23 for ≥65 year olds. The rate increased from 0% in 2009 to 10% in 2011, 40.6% at the end of 2015 and 74% at the end of 2018. After PCV7 and PCV13 were implemented in the child vaccination programme change in serotype prevalence included an increase in serotypes 8, 9N and 12F which comprise 40% of the serotypes which cause IPD in the elderly but which are included in PPV23 and thus provided them with protection [25]. In Japan a postal and web-based nationwide survey was sent to all municipalities (*n* = 1741) in June 2016 and 1010 municipalities (58.0%) responded to the survey. Median PPV23 coverage for adults aged 65 years for responding municipalities in 2016 was 41.8% and was higher by 18.7% (16.7%–20.7%) in the municipalities which sent direct mail notification to targeted adults. It decreased by 3.02% (2.4%–3.6%) for every 1000 Japanese Yen increase in out-of-pocket costs to individuals, and PPV23 coverage was inversely related to municipality unemployment rates and average per capita income [26]. In England, the coverage of PPV23 in those ≥65 was similar to Japan and was 70.1% in 2015 and 69.5% in 2018 [27]. In Australia after public funding for PPV23 commenced in 2005 the vaccination rate increased from 35.4% pre-2005 to 56.0% after 2005 [28]. In South Korea during a 20 month national immunisation program the pneumococcal vaccine rate for ≥65 years increased from 5.0% to 57.3% [29].

### 3.4. Detection of Pneumococcal Pathogens in Adults Hospitalised for CAP

Detection of pneumococcal pneumonia initially depends on clinical symptoms and new infiltrates on chest Xrays (CXR) and then varies according to the laboratory tests used. A prospective surveillance study in 21 acute care hospitals in the US 2013–2016 demonstrated how identification of pneumococci as pathogens was affected by the techniques used. Individuals ≥18 presenting to emergency departments with suspected pneumonia were screened by clinical findings (fever or hypothermia within 24 h of enrollment, chills or rigors, pleuritic chest pain, cough, sputum production, dyspnea, tachypnea, malaise, and abnormal auscultatory findings suggestive of pneumonia) and of a total of 15,572 individuals admitted with CAP, 12,055 had a CXR interpreted as CAP and constituted the final analysis sample. *S. pneumoniae* was detected in 122 (1%) by culture, in 345 (2.9%) by a serotype-specific urine test, and in 447 (3.7%) by BinaxNOW for pneumococcal C-polysaccharide urinary antigen detection (UAD). By any of the three tests the detection rate was 10.7% in those 18–64 and 9.2% in those ≥65. Thus multiple methods of detection improve the *Streptococcus pneumoniae* detection rate, but most diagnoses still depended on clinical findings and a CXR obtained no more than 72 h prior to enrollment suggestive of pneumonia. Mortality within 30 days of hospitalisation for those 18–64 at low risk was 0%, for those at high risk 7.6% and for those ≥65 was 11.3% [30].

A study of CAP in two university hospitals in Nottingham, UK 2008–2013 showed higher rates of detection with laboratory methods. Patients ≥16 years were eligible for study inclusion if they had at least one symptom of lower respiratory tract infection (cough, increasing breathlessness, sputum production or fever) and were treated by their clinical team for CAP (*n* = 2702). After excluding those who did not consent or had an alternative diagnosis, 2224 constituted the sample with a median age of 71 (IQR 56–80). For the 643 CAP patient diagnosis was based on either a positive pneumococcal UAD (8.7% of patients did not provide a urine sample), or a positive blood culture for *S. pneumoniae* or pneumococcal serotype detection by Bio-plex assay and one or more CAP serotypes were diagnosed in 66.7% of cases with these tests and the remainder remained untyped using these methods. The incidence of pneumococcal CAP was 64/100,000 in those ≥65 and 20.7/100,000 for the entire cohort. For patients ≥65 chronic heart disease (29%) and COPD (28%) were the most common risk factors. The 30 day mortality rate for those ≥65 with a clinical risk factor was 14.2% and without a clinical risk factor 8.6% and for those <65 with a clinical risk factor 1.5%. Thus the pneumococcal detection rates in CAP patients were much higher than in the US study by Isturiz and generalisations based on specific hospital catchment areas need to be made cautiously [31].

A systematic review and meta-analysis of 33 observational studies 1980–2015 of the effectiveness of PPV23 in preventing CAP requiring hospitalisation in those ≥65 found VE = 10.2% (−12.6, 33.0) and the authors cautioned that the data were from a wide diversity of populations, circulation of *S. pneumoniae* serotypes, pediatric pneumoccocal vaccination coverage, case definitions and time since vaccination [32].

### 3.5. Effect of Vaccination on Antibiotic Resistance to Medications Used for Pneumococcal Disease

An important question is whether the introduction of pneumococcal vaccines reduced infection rates, and also had benefits in reducing antibiotic use and consequently reducing antibiotic resistance rates. A systematic review 2008–2017 of 32 studies on the relationship of the introduction of PCV7, PCV10 and PCV13 and subsequent pneumococcal carriage rates and IPD resistant to antibiotics found that 12 studies found significant declines (20–76%) in IPD resistant to penicillin and 10 no change; 10 studies found significant declines (11–100%) in IPD resistance to cephalosporins and 8 found no change; and 6 studies found significant declines (19–53%) in resistance to macrolides and three found no change. The changes were mostly observed in countries with the highly prevalent antimicrobial-resistant serotypes 7F 19A before the introduction of PCV13 (France, Japan, South Africa, Turkey and the US). After the introduction of PCV13 there was also a significant decrease in the use of antibiotics in multiple countries for otitis media but an increase in the serotypes 8, 12F, 15B/C, 22F and 33F not covered by PCV13. The serotypes covered by PCV are also the ones most associated with antibiotic resistance and reducing the occurrence of these would reduce resistance levels to the antibiotics usually prescribed Thus important strategies for ensuring better outcomes for PP and IPD are both prompt diagnosis and antibiotic treatment for PP and IPD not prevented by vaccination and reducing the prevalence of antibiotic-resistant pneumococcal disease by improving infectious disease antibiotic stewardship [33].

### 3.6. Effect of Vaccination on IPD Caused by Antibiotic-Resistant Pneumococcus

A systematic review of 13 RCTs and 86 observational studies of the effect of PCV vaccines on serotype 19A nasopharyngeal carriage concluded that in countries with relatively high pneumococcal antibiotic resistance using higher-valent PCV vaccines for routine childhood immunisation reduced antibiotic resistant IPD, otitis media and nasopharyngeal carriage rates in children and also antibiotic-resistant IPD in adults and that the effectiveness of PCV13 against serotype 19A was an important factor [34].

A systematic review 1946–2017 of randomised controlled trials and observational studies of the incidence of IPD, non-invasive pneumococcal disease, hospitalisation and mortality in adults before and after the introduction of PCV13 for children found that there were significantly lower IPD incidence rate ratios in both adults <65 years (IRR 0.78; 0.72, 0.85) and ≥65 years (IRR 0.86; 81, 0.91) but there was a non-significant change in IPD due to non-vaccine serotypes for those <65 IRR 1.04 (0.95, 1.14) and a significant increase in those ≥65 (IRR 1.20; 1.11, 1.29). However, after PCV13 was introduced for children the IRR for PP in adults for those <65 was 1.05 (0.82, 1.35) and for those ≥65 was 0.99 (0.84, 1.16) [35].

PCV7 was introduced in The Netherlands in 2006 and PCV10 in 2011. A national IPD surveillance study in The Netherlands from 2004 to 2018 reported data from sentinel laboratories in 2004 and 2018 covering ~25% of the population. Compared to before the introduction of PCV7, in 2016–2018 IPD incidence declined in children <5 years old by 69%, in 5–17 year olds by 40%, in 18–49 year olds by 31% and in ≥65 year olds by 19%. Compared to before PCV10 was introduced IPD incidence 2016–2018 declined in children <5 years old (RR 0.68; 0.42–1.11), in 5–17 year olds (RR 0.58; 0.29–1.14), in 18–49 year olds (RR 0.72; 0.57–0.90), but there were no significant changes in 50–64 year olds (RR 0.94; 0.81–1.10) or ≥65 year olds (RR 1.04; 0.93–1.15). The case fatality rate was 16.2% before PCV7, 11.8% after PCV7 and 13.4% post PCV10. For those ≥65 post-PCV10 compared to pre PCV7 RR was 0.75 (0.62, 0.90) and for post-PCV10 compared to post-PCV7 RR was 1.10 (0.91,1.34), likely due to serotype replacement [36].

### 3.7. The Effect of Pneumococcal Vaccination on Carriage Rates in Those ≥65

There are substantial and varying pneumococcal nasal and pharyngeal carriage rates in all age groups, and an issue is whether the use of pneumococcal vaccines changes those carriage rates and selects for non-vaccine serotypes. In the CAPiTA RCT in The Netherlands for 2011 individuals ≥65 (average age 72.5) the pneumococcal carriage rate measured by PCR *lytA* was 17.9% (of which 6.2% were nasopharyngeal and 14.8% oropharyngeal carriage). Patients were randomised to receive either PCV13 or placebo and six months after vaccination pneumococcal carriage rates were 19.8% (17.3, 22.6) for the placebo and 15.3% (13.1, 17.8) for the PCV13 vaccinated group RR 0.77 (0.83, 0.94). At 24 months post-vaccination pneumococcal carriage rates were 18.8% (16.2, 21.5) for the placebo and 16.4% (14.1, 19.08) for the PCV13 vaccinated group RR = 0.88 (0.72, 1.07), with similar nasal and oropharyngeal percentages at each time point. The differences between the groups at any time for any vaccine-treatable serotypes were ≤1.1%, for any nonvaccine serotype ≤1.5%, and for any untypable serotype ≤0.4%. Thus PCV13 vaccination in this study changed neither nasopharyngeal carriage rates nor replaced vaccine-treatable serotypes with non-vaccine serotypes [37].

A large gene study in the south-west of the US illustrates how the introduction of vaccines affects the reduction in vaccine-treatable serotypes and the increase in non-vaccine serotypes. Navaho and White Mountain Apache communities before pneumococcal vaccines were introduced had IPD rates 2–5 times higher than the general population and 50% carriage rates (and 75% carriage rates among children <2 years). Three prospective studies 1998–2012 provided 812 pneumococcal samples from children ≤5 years and 125 from individuals 6–76 years of age. The samples from 1998–2001 were from vaccine-naïve populations, from 2006–2008 from groups which received PCV7 and from 2010–2012 were from groups that received the PCV13 vaccine. From these groups 8674 clusters of orthologous genes (COGs) were identified, and 1111 were present in >99.9% of strains and formed the core genome. The pneumococcal capsule is the strongest determinant of virulence and predictor of prevalence and at the time of the study 93 capsular serotypes were known. Pneumococcal vaccination changed both the prevalence of serotypes and the pneumococcal pangenome. The pangenome includes capsular antigens, noncapsular antigens and mobile genetic elements which affect virulence and transmission and thus the overall ecology of the pneumococcus in the communities in which pneumococci are transmitted. After the introduction in 2000 of PCV7 IPD rates decreased by 89% and the pangenome in these populations became smaller. However, after PCV7 use the frequencies of the genes coding for host-pathogen interactions which were initially disrupted returned to pre-vaccine values. There were also increases in numbers both of serotypes not treatable by PCV7 and serotypes previously unobserved. The initial frequencies of genes that coded for proteins involved in host-pathogen interactions decreased but then returned to pre-vaccine values. [38].

### 3.8. Persistence of Immunity Following Conjugate and Polysaccharide Vaccination in Seniors

A key issue is how frequently pneumococcal vaccine boosters need to be given, and for which age groups and for which groups with risk factors. Studies have measured IgG and opsonophagocytic antibody levels and B cell levels.

The study with the longest follow-up period (six years) is an open label RCT of 312 unvaccinated hospitalised patients ≥60 years which randomised them to either PPV23 or to PCV7 followed by PPV23 six months later. Self-report of vaccination was validated by each participant’s physician. At the six year follow up of the 215 surviving participants 136 were reassessed and in the PCV7 group Geometric Mean Concentrations of IgG levels measured by ELISA persisted above baseline levels for serotypes 3, 4, 6A, 6B, 9V, 14, 18C, 19A, 19F and 23F and in the PCV7-PPV23 group for 3, 6A, 6B, 9V, 14, 19F and 23 F. For both study arms Geometric Mean Titres (GMT) of opsonophagocytic antibodies (OPA) levels decreased to or below baseline levels for serotypes 4, 6A, 6B, 9V and 23F and remained above baseline levels for 14, 18C and 19A. ELISA and OPA titres were significantly correlated for all serotypes. In the PCV7-PPV23 arm GMT levels of OPA were significantly higher for serotypes 18C and 23F for participants with a low frailty index compared to those with a moderate or high frailty index. Overall 72 months after baseline there were no significant differences in geometric mean concentrations (GMC) of IgG or OPA between study arms for all tested serotypes and there were no significant differences in readmission or death between the two study groups [39].

The RCT with the next longest follow up period (five years) is an RCT which randomised 1008 healthy individuals >50 in the US to PPV23 or placebo. Baseline antibody levels for those who had previously received PPV23 vaccine were 2–3 times higher than for pneumococcal vaccine-naïve subjects but within each of the vaccine-naïve and previously vaccinated groups were similar for the age groups 50–64 and ≥65. For the 564 present after five years (58% of the previously vaccinated and 96% of the primary vaccination groups) the pneumococcal GMC IgG levels for previously vaccinated subjects were higher than at baseline except for serotype 3 and remained ≥2 fold greater than for the vaccine naïve group [40].

The study with the next longest follow up period (1 year) is a double-blind RCT in 46 centres in the US and seven centres in Sweden of 936 adults ≥70 who had received PPV23 at least five years earlier and in which the subjects were randomised to PCV13 or PPV23. The power computation required 462 participants per study group for 90% power to assess PCV13 as noninferior to PPV23 for the 12 pneumococcal antigens common to both vaccines, using a 2-fold noninferiority criterion of 0.5, a 2-sided type 1 error rate of 0.05 and a drop-out rate ≤12%. One year later 795 were assessed and the OPA GMTs in the PCV13 group were significantly greater for 10 of the 12 serotypes common to both vaccines and also for serotype 6A. Both groups were then vaccinated with PCV13 open label and 745 patients were followed up within six months after this final vaccination. For the vaccine sequences a year apart for the PCV13/PCV13 arm GMT OPA levels for 11 of 12 common serotypes and 6A were statistically significantly higher than for the PPV23/PCV13 arm whereas there were no statistically significant differences for the sequences PPV23/PCV13 compared to PCV13/PCV13. For the age groups 70–74, 75–79 and 80+ the responses after PCV13/PCV13 were greater than for PPV23/PCV13 for the majority of serotypes for those 70–74 and 75–79 and slightly lower in those 80+. The PCV13 serotypes are responsible for a substantial worldwide burden of pneumococcal disease in adults and thus these results for the older age groups within the group ≥65 are important [12].

Two studies assessed results after different sequences of PCV and PPV23 vaccines. An RCT randomised 348 individuals 50–70 either to PCV7-PCV7-PPV23 or PPV23-PCV7-PCV7 or PCV7-PPV23-PCV7 with vaccines administered six months apart. Prior to pneumococcal vaccination 86% of participants had detectable memory B-cells that secreted antibody to at least one of the seven serotypes tested. For those who received PCV7 28 days later there was an increase in memory B-cells but for those who received PPV23 there was a decrease. For those who received PCV7 six months after PPV23 the responses of memory B-cells (which affect T-cell response) were lower than after a single of dose of PCV7 but recovered after a second dose of PCV7. If individuals received one or two doses of PCV7 and then 6 months later received PPV23 their memory B-cell response was lower. Thus because PCV induces a T-cell response but PPV23 does not, PCV7 may have an immunological advantage in this age group [41,42].

An open-label RCT in Germany randomized 443 healthy community dwelling ambulatory adults ≥70 (average 75.4 years, range 69–90) who had no previous history of pneumococcal vaccination to vaccination in four arms: either PCV7, or PCV7 and PCV9, or 2 doses of PCV 7 and 2 doses of PCV9, or PPV23. If patients received a second vaccine it was one year later but assessment was only one month after the second vaccination. Baseline OPA responses were similar among the four groups with the highest level for serotypes 9V (110.7–173.1) and 14 (64.0–116.5) and lowest for 4 (13.6–16.3) and 19F (12.6–16.3). One month after final vaccination GMT OPA levels for the three PCV arms were higher for all serotypes than for PPV23 except for 19F and there were no statistically significant differences between the PCV arms [43].

One small RCT compared responses after PCV and PPV23 but only assessed patients one month after vaccination. It included some of the oldest patients whose ages were reported. From five nursing homes around Tokyo 623 eligible subjects were identified and 105 residents over 80 (average age 88; 45 were 90–100 and three 101) with no prior pneumococcal vaccination received PCV7 or PPV23 vaccination and in the PCV7 group there were significant elevations in both IgG and OPA levels for serotypes 4, 9V, 18C, and 23F compared to the PPV23 group but not in 6B, 14, or 19F [44].

### 3.9. Effect of Pneumococcal Vaccination on the Incidence of Pneumococcal Pneumonia, Invasive Pneumococcal Disease, Hospitalization and Death

The study with the longest follow-up (7.3 years) is the Kaiser Permanente California Men’s Health Study cohort study of 39,222 men 45–69 of whom 9910 terminated membership and 1313 died before the end of the study, leaving 29,312 to be assessed. Health status and diseases were assessed only by ICD-9 codes and microbiology reports in their electronic medical records. The Cox regression models controlled for age, race/ethnicity, region, household income, education, body mass index, cigarette smoking, physical activity level, sedentary behaviours >6.5 h daily outside work, alcohol consumption, number of influenza vaccinations received, total calorie intake, fat intake, fruit and vegetable consumption, heart disease, cardiovascular heart disease, diabetes, COPD, kidney disease, cancer, liver disease, dementia and HIV status. Thirty per cent had received PPV23 before the baseline and 20% their first PPV23 after the baseline. Those who received PPV23 after age 65 had lower smoking rates, very good or good health and engaged in more vigorous activities, indicating PPV23 had been prescribed more frequently to those when <65 who had adverse health conditions and demonstrates the importance of controlling for comorbidities in cohort studies. Low rates of bacteremia were detected (*n* = 17) of which 9 were in the unvaccinated. The number of all-cause pneumonia hospitalisations in the unvaccinated was 177 (0.9%); for those who received PPV23 before age 65, 250 (3.0%); after age 65, 158 (1.8%) and both before and after 65, 62 (2.1%). The findings of no benefit from the vaccine and low numbers of bacteremic cases (17) may be due to the method which used only medical records and laboratory reports [45].

Two studies assessed patients and recorded pneumococcal cases as they accrued during the pre-specified study period but did not report the average or range of follow-up times after vaccination. The CAPiTA (Community-Acquired Pneumonia Immunization Trial) of 84,946 adults ≥65 in The Netherlands with no prior pneumococcal vaccination history randomised participants to PCV 13 or placebo. Recruitment was between 15 September 2008 and 30 January 2010 and evaluated cases of “suspected pneumonia” and IPD acquired between 15 September 2008 and 28 August 2013. After 130 cases of CAP were detected assessment at 59 sentinel centres was concluded. The average or range of follow-up periods was not stated. Twenty-five per cent had heart disease, 10% lung disease, 13% diabetes, 88% were non-smokers and baseline characteristics were similar between the groups. CAP was assessed clinically by ≥2 prespecified clinical criteria, radiographic confirmation, culture and serotyping from blood and sterile sites (79%) and nonsterile sites (47%), PCV13 serotype-specific UAD assay (93% of participants), a proprietary Pfizer multiplex antigen-binding assay for all PCV13 capsular polysaccharide serotypes, BinaxNOW a pneumococcal C-polysaccharide assay (94%), and gram staining and microscopy of non-sterile samples (48%). The modified intention to treat (ITT) analysis was specified for participants with an episode of CAP at least 13 days after vaccination and of the 23 pre-specified endpoints VEs for 10 were significant. For the PCV 13 vaccine VE for all episodes of culture-confirmed CAP (73 cases) was 51% (19%, 71%), for all episodes of culture-confirmed vaccine-type pneumococcal CAP (29 cases) was 74% (34%, 91%), for all episodes of culture-confirmed non-vaccine treatable community acquired pneumonia (NVT-CAP) (44 cases) was 31% (−31%, 64%); and for all episodes of non-vaccine treatable invasive pneumococcal disease (NVT-IPD) (41 cases) was 24% (−31%, 57%). Thus vaccination confers major benefit for culture-confirmed vaccine-treatable community acquired pneumonia (VT-CAP) but the outcomes for NVT-IPD include unity and are non-significant [46].

A cohort study evaluated 27,204 community dwelling individuals ≥60 from nine primary care centres in Tarragona, Spain for the outcomes of PPV23 vaccination. CAP was defined as an acute respiratory illness with new infiltrates on a CXR, and bacteremic CAP when *S. pneumoniae* was isolated from blood or other sterile fluids. The clinical investigator who validated clinical data and the microbiologist were blinded. The rate of influenza vaccination rate in the previous autumn was much higher in the vaccinated group (82%) than the never vaccinated (16%) group. For patients who received PPV23 within the previous five years compared to the never vaccinated hazard ratios (HR) were adjusted for age, sex, influenza vaccination, nursing home residence, COPD, diabetes and cancer. For bacteremic CAP (12 cases) HR = 0.38 (0.09, 1.68; *p* = 0.203) for overall pneumococcal CAP (84 cases), HR = 0.49 (0.29, 0.84; *p* = 0.009) and for all-cause CAP (375 cases) HR = 0.75 (0.58, 0.98; *p* = 0.033) whereas for all patients who had received PPV23 including more than five years earlier none of the outcomes were statistically significant. This illustrates both the problems of identifying a sufficient number of laboratory-proven cases of pneumococcal CAP, the requirement of large samples, and the waning of PPV23 vaccination effectiveness after 3−5 years [47].

An RCT in Japan randomized 1006 nursing home residents in 23 nursing homes to PPV23 or placebo and they were followed for an average of 2.3 years. Pneumonia was diagnosed clinically and by a new infiltrate on CXR (checked independently by another reader) and in 70% of cases was confirmed by Computer Assisted Tomography (CAT) of the chest. Pneumococcal pneumonia was diagnosed by two samples of blood, pleural fluid or sputum for culture or by the pneumococcal antigen BinaxNOW urine detection kit. The RCT was triple blinded (patients, evaluators and statistician).

The power computation was based on the pneumococcal pneumonia rate in nursing homes in Japan and for power of 80% and statistical significance <5% required a follow up period of three years with 700 participants, a placebo group of 350 participants and 43 cases of pneumococcal pneumonia. During the follow up period all-cause pneumonia was diagnosed in 167 patients (12.5% in the vaccine and 20.9% in the placebo group) but a full set of tests was carried out in only 58% of the 167. Forty-nine (29%) were diagnosed with pneumococcal pneumonia (49 by urinary antigen tests, 41 by sputum culture and three by blood culture) and other causative pathogens were *Staphylococcus aureus* (6%), *Enterobacteriaceae* (5%) *Haemophilus influenzae* (3%) and *Pseudomonas aeruginosa* (2%). Pneumococcal pneumonia was diagnosed in 2.8% of the vaccinated and 7.3% of the placebo group. In the vaccine group 13/63 died from all-cause pneumonia and 0/14 from pneumococcal pneumonia and in the placebo group 26/104 and 13/37 respectively [48].

Authors from Pfizer critiqued Maruyama’s study on the basis of improbable VE estimates, no standard definition of radiological diagnosis of CAP and uncertainty as to whether the allocation by random number table by the nurses was of individuals or groups of patients.

“Maruyama et al. reported that 31% (51 of 167) of pneumonia cases were due to pneumococcus with a VE against pneumococcal pneumonia of 64% (Table 1). To achieve this VE against pneumococcal pneumonia, one would need to assume that 64% of pneumococcal pneumonia was due to PPV23 serotypes with a PPV23 VE of 100% against these vaccine serotypes; that 100% of pneumococcal pneumonia was due to PPV23 serotypes with a PPV23 VE of 64% against these vaccine serotypes; or some combination of values between these extremes.” [49]. The effect of including the Maruyama study as an outlier in meta-analyses of VE and in guidelines has been noted [50].

### 3.10. Studies Which Performed Analyses Controlling for Risk Factors

Two studies have documented the serious influence of chronic disease risk factors in seniors on the incidence and outcomes of pneumococcal diseases. In Taiwan a free nationwide PPV23 vaccination programme for those ≥75 was conducted from October until December 2008. A study using the Taiwan National Health Insurance Research Database (NHIRD) and the Ministry of Health invasive pneumococcal disease database identified 1,078,955 eligibles, of whom 318,257 (29.5%) received PPV23 and these were matched by propensity score matching on age, gender, influenza vaccination status, chronic diseases and related care utilization to a sample which was not vaccinated. After propensity score matching for the purposes of analysis the influenza vaccination rate was 83% in both groups. Vaccinees had lower ORs for IPD 0.24 (0.123, 0.461), death from IPD 0.09 (0.011, 0.704); hospitalisation for pneumonia 0.40 (0.395, 0.415), death from pneumonia 0.07 (0.059, 0.082), and all-cause mortality 0.07 (0.069, 0.072), all *p* < 0.001. Of the vaccinees 271,806 (85%) and 203,298 (27%) of the non-vaccinees also received influenza vaccination in 2008. Thus the PPV23 vaccinee group had a 76% reduction in the risk of IPD and a 91% reduction in the risk of PP. The study did not report on age subgroups within this population of individuals ≥75 [51].

In the CAPiTA double blind RCT of 84,496 immunocompetent adults ≥65 49% self-identified themselves as at-risk due to chronic medical conditions (heart disease, lung disease, asthma, diabetes, liver disease, smoking and a history of splenectomy) and were followed for an average of four years. There were 139 VT-CAP cases of which 115 (83%) occurred in the at-risk group for a rate of 90.6/100,000 compared to 21.1/100,000 in the not at-risk group. For those who had received PCV13 the modified intention-to-treat population efficacy analysis for VE was 43% (11.4%, 60.2%) [52].

### 3.11. Studies Which Performed Stratified Analyses by Age Groups 65–74, 75–84, 85–89, and 90+

One RCT provides data on how VE declines in the older seniors above 80 compared to the younger seniors. The CAPiTA trial has the largest numbers in the age groups above 65: 32,933 <70 years, 25,145 (70–74), 15,758 (75–79), 7715 (80–84) and 2941 (85+). The average follow-up was 3.9 years and during follow up 6011 (7.1%) subjects died and 4572 (5.4%) were lost to follow-up. There were 184 first episodes of VT-CAP or VT-IPD. Blood cultures were performed in 78% and urine for serotype specific UAD in 93% of those <70 years and 90% >85 years. VE declined from 65% (38%, 81%) in 65 year olds to 40% (17%, 56%) in 75 year olds and close to zero for the 83 year olds (no numbers provided, but by inspection of the point estimate in Figure 1 in the article), and were similar for VT-CAP and VT-IPD [53].

## 4. Discussion

Pneumococcal pneumonia (PP) and invasive pneumococcal disease (IPD) are important causes of morbidity and mortality at all ages but especially in seniors and children. The introduction of pneumococcal conjugate vaccines (PCVs) for children from 2000 had a dramatic effect via herd protection on reducing the circulating pneumococcal serotypes most responsible for causing PP and IPD in seniors.

Diagnosing PP and IPD depends initially on clinical symptoms (fever or hypothermia, chills or rigors, pleuritic chest pain, cough, sputum production, dyspnea, tachypnea, malaise, and abnormal auscultatory findings suggestive of pneumonia), then on a CXR with new infiltrates. Confirmation is by UAD, sputum or blood cultures but a substantial number of cases, depending on the patient population tested, may depend on clinical symptoms and CXR because bacteremia is not ascertained in many pneumococcal pneumonias.

Key factors in higher rates of PP and IPD and mortality in seniors are declining immunity and a substantial burden of risk factors due to multiple chronic diseases especially as they become older seniors. There are data on vaccine effectiveness (VE) for those ≥65. Falkenhorst’s (2017) systematic review identified four RCTs and thirteen observational studies (five cohort studies, three case control studies and five case studies). For IPD for the RCTs the pooled vaccine effectiveness (VE) was 73% (95%CI 10%, 90%; I^2^ = 11%) and for the cohort studies at low risk of bias, VE was 45% (15%, 65%, I^2^ = 0%). For PP for the two RCTs rated as low risk for this outcome VE was 64% (35%, 90%; I^2^ = 0%) and for the cohort studies, 48% (25%, 63%, I^2^ = 0%). The GRADE quality of evidence was moderate for the RCTs [9].

However, there are minimal data for VE for those ≥65 for the subgroups by age deciles and no data by age quintiles for 70–74, 75–79, 80–84, 85–89 and 90+. Inspection of any demographic data listing population numbers in these age groups shows the rapid diminution in numbers as each age decile or quintile is reached and thus data on pneumococcal rates, risk factors, pneumococcal carriage rates, current serotypes and pneumococcal antibody are vital for maximising the life expectancy of older seniors.

Vaccines reduce the frequency of vaccine-treatable serotypes in the population and these serotypes tend to be replaced by both non-vaccine treatable serotypes and serotypes not previously widely circulating. Studies have tested different sequences of PCV and PPV23 vaccines. Individuals who had been vaccinated before the trial had higher antibody levels than vaccine-naïve participants. Patients who received PCV tended to have higher IgG and opsonophagocytic antibody levels and B cell levels against the vaccine-treatable serotypes than those who received PPV23.

Administering PPV23 vaccines to seniors made minimal changes in their nasal and pharyngeal carriage rates and there is thus a steady supply of commensal serotypes available to become invasive. There are no studies of PP or IPD or pneumococcal vaccination rates or carriage rates in the family members or carers of seniors and thus it is not known to what extent seniors contract PP or IPD from these sources. There are no studies of personal distancing, handwashing, mask wearing or pneumococcal and influenza vaccination rates among the carers and family members of seniors in nursing homes or among those seniors who get IPD or PP.

For seniors in long term care and nursing homes an important innovation will be changing air flows so that all bedrooms, dining and meeting rooms and corridors vent to the outside and air locks in corridors reduce transmission of respiratory infections. Some seniors spend up to 22 h a day in their rooms and thus air flow from the outside descending from the ceiling onto the area containing their bed and easy chair then led outside will reduce transmission. Studies of influenza shedding are relevant to reducing the transmission of all respiratory diseases. A study evaluated influenza shedding by 61 patients in a North Carolina hospital in rooms with six air changes/hour, at 20 °C, relative humidity 40% and end filters compliant with American National Standards Institute standard 52.2–2007. A foot from the patient’s head 300 RNA copies >4.7/m and 100 RNA copies ≤4.7/m were detected, with the opposite particle size distribution six feet from the patient’s head (5 RNA copies >4.7/m and 80 RNA copies ≤4.7/m). The five highest emitters shed 32 more times virus (up to 20,400 RNA copies per 20 min) compared to the other emitters (<1300 RNA copies) [54].

Super shedders were also identified in a study in Hong Kong. Twenty per cent of the most infectious children with influenza were responsible for 96% of total viral shedding by children (average influenza viruses shed/infection = 9 million (range for the 20th to 80th percentile = 800,000 to 100,000,000), and 20% of the most infectious adults were responsible for 82% of the total adult viral shedding (average shed/infection = 20,000,000 (range for the 20th to 80th percentile = 4,000,000 to 90,000,000) [55].

The Schools of Architecture and Engineering at the University of Cambridge have designed air flows for temporary large COVID-19 wards and have demonstrated how air flows can be changed to minimise respiratory pathogen transmission [56,57]. These innovations can be adapted for nursing homes and long term care facilities

## 5. Conclusions

To understand how seniors and especially older seniors acquire PP and IPD data are needed on community transmission from visitors, family members, carers and community contacts. For all nursing homes and long-term care facilities data are needed on current influenza, COVID-19 and pneumococcal disease rates, influenza, COVID-19 and pneumococcal carriage rates for all patients in these age groups, staff members, family members and other visitors and new virus variants. Monitoring of sewage outflows in communities in The Netherlands provided warning of COVID-19 presence in the community before residents showed any symptoms or signs and sewage monitoring needs to be implemented to pinpoint the presence of infected residents and staff. The medical and nursing direction of each home can then produce detailed plans for ongoing surveillance of staff, seniors, family members and visitors and how to provide nursing and personal care to minimise transmission. Ideally new nursing homes need to be built as small separate units with dedicated nursing and care staff for each unit to minimise respiratory disease transmission. Transmission from other services such as personal care, food, toileting, laundry and maintenance will also need to be minimised.

National programmes to increase pneumococcal and influenza vaccination rates for seniors have been shown in multiple countries to be effective in increasing rates but not in maximising these rates [58]. Supervision is needed at the national level to increase programme effectiveness to achieve the current US and European goals, especially for those groups clearly identified by multiple studies as having lower vaccination rates and also being at higher risk of PP and IPD.

In nursing homes the patients, family members, medical, nursing staff and carers need to be fully vaccinated.

Chronic diseases (heart, lung, renal, and liver disease, poor immune competence and dementia) are risk factors for poor outcomes from PP and IPD and need to be optimally managed to maximise benefits from vaccination in avoiding PP and IPD. Nursing homes need redesigning to enable residents to live in a room they do not share with others and in small separate building units to minimise respiratory infections. Each small building should have its own dedicated staff of carers. Food should be individually prepared in each building and laundry performed there. Careful monitoring of staff and family members for infection, handwashing and respiratory precautions during respiratory seasons should be key preventive manoevres. Staff and family members should be fully vaccinated against influenza, pneumococcal disease, and corona viruses. Existing large nursing homes need to implement as many of the above strategies as they can and study the airflows in the home to ensure the air flow from each room and corridor vents to the outside and airlocks separate the buildings into small units.

## Data Availability

Data are all from published articles referenced in the text and listed in the list of References.

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
