# Peer review of "Pneumococcal Pneumonia and Invasive Pneumococcal Disease in Those 65 and Older: Rates of Detection, Risk Factors, Vaccine Effectiveness, Hospitalisation and Mortality"

_geriatrics, 2021, doi:10.3390/geriatrics6010013_

Round 1
Reviewer 1 Report
The current article is a review of the literature to determine the effect of various vaccination schedules for patients over the age of 65. Overall the article is well written and breaks down the presentation of data into four main points that need to be addressed when determining the role of vaccinations on health in the indicted age groups. The author did a sufficient literature review to address the main points and indicated relevant findings from the literature cited. There is mainly in editing of the writing. Detailed below are issues found with writing that hinders comprehension of the points being raised and that should be addressed. In general a careful proofing of the article is also needed.
Ln 40-42. The first point made is hard to follow, especially when mentioning the introduction of the various PCV vaccines. There is a lot going on in this sentence and needs to be broken up.
Ln. 49. An unnecessary period after “seasons”
Ln 81. RCT should be defined
Ln 302-306. Hard to follow. More detail is needed to aid in understanding, particularly when talking about "duration of PCV7 exposure." Is this referring to longer colonization periods of PCV serotypes? Clarity is needed.
Ln 311. How did the pneumococcus adapt? Did the same core gene that were reduced after vaccination return or were there different genes that were termed core genes, but numbers of core genes the same. Another sentence explaining the change in gene content would be appropriate here.
Ln 319. An underscore after “physician” is not needed.
Ln 322. A space after 23 in the 23F serotype is added unnecessarily.
Ln 333. Sentence needs to be restructured. “above maintained above”. Meaning is lost in this sentence.
Ln 333. “blevels” unsure if this refers to baseline levels or if misspelled. If intentional should be defined.
Ln 333. “. seline” unclear what this refers to.
Ln 349. “naive” needs to be naïve
Ln 374. “there was” is superscript
Ln 379. Add a comma after “does not”
Ln 465. There is no closing parenthesis for the “(“ after “CXR”
Ln 506. Add a 0 before period in “ p <.001.” so p <0.001.
Reviewer 2 Report
This review is a good supplement to the presentation of data within pneumonia and IPD in the elderly, and how it affects the different age groups from 65+. However, I have some comments and suggestions.
- In general, it is difficult to obtain an overview of the included references. I strongly suggest that the authors include either a flowchart or a table to present the articles, which have been used in each topic.
- Regarding the pneumococcal vaccines, it needs to be clearer in the review which specific vaccine the authors refer to in each specific section. For example in section 3.3. “Pneumococcal vaccination rate”, it is not clear which vaccines the data are referring to.
- At lines 38 – 43, the authors mention the PCV vaccines with regard to the elderly. To my knowledge, PCV-7 and PCV-10 were never approved for use in persons older than 5 years of age. PCV-13 and PPV-23 are the only pneumococcal vaccines that to my knowledge have been licensed for senior citizens. I think you need to either add some reference or correct the sentence!
- At line 49, “disease seasons.by increasing” does not make sense!
- Lines 86 – 87: The sentence “The pneumococcal vaccines are abbreviated differently between publications and are designated here as PPV23 and PCV7, PCV10, and PCV13”, is repeated at lines 107 – 108!
- Several places in the manuscript “herd immunity” is mentioned. I think that it is more correct to use “herd protection”!
- Line 110: I think “then” should be replaced with “than”!
- At lines 187 – 188, the authors describe the PPV23 coverage based on ref 19, a Japanese study. I think it would be more correct also to include an English study, for example Djennad et al 2018 (EClinicalMedicine 6 (2018) 42–50).
- This topic of this review is pneumonia and IPD. I think the review lacks a clearer description of how pneumonia/IPD is clinical defined, e.g., the difference between non-invasive pneumonia and invasive pneumonia! Either a table or a figure could be useful. The description furthermore needs to be in the first section of the manuscript, and not in the discussion, lines 535 – 540.
- Regarding discussion of PCV7, PCV10, and PCV13, especially with regard to serotype 19A, I think the review would benefit from including the study by Desmet et al 2021 (DOI: 1016/S1473-3099(20)30173-0), on the Belgian experience of PCV7 versus PCV10, versus PCV13!
- Lines 292 – 311: the topic of this section seems irrelevant to the general topic of section 3.7. I suggest that the authors remove it or completely rephrase the section!
- Line 333 does not make sense!
- Lines 367 – 368, the phrase “older old age groups”: I think you need to specify the age groups!
- Line 370, the sentence “An RCT randomised 348 individuals 50-70 to”: you need to add years of age!
- At line 374, there is a typo problem!
- At line 139 S. pneumoniae should be the full name (Streptococcus pneumoniae), while at line 449 it should be S. pneumoniae!
- Lines 560 – 564 need to be rephrased: I do not understand this sentence!
- In the reference list, there is a problem with references 9, 10, and 11, which need to be fixed!
Author Response
Please see attachd Word file

Reviewer 3 Report
The manuscript is of relevance as it gathers and highlights data available on the burden of pneumococcal diseases and pneumococcal vaccination effects in the elderly population. However, besides comments and notes made throughout the manuscript (attached), three main issues should be highlighted:
- About the title of the manuscript: I suggest removing the stratified ages and refer to them as older than 65 since there not so many studies that present results stratified by ages (as also stated in the manuscript conclusion). Also I suggest removing the word "carriage" as it refers to an asymptomatic condition and not is necessarily linked to pneumococcal disease.
- There are some data and affirmatives present throughout the manuscript that need to be referenced.
- In general, the text is composed by sentences that are too long and/or missing punctuation, and, thus, confusing. I strongly suggest revising this aspect.

Round 2
Reviewer 2 Report
The manuscript has improved, however I still have some comments and suggestions.
- I still find it is difficult to obtain an overview of the included references, and therefore feel the manuscript would benefit, if the authors included either a flowchart or a table to present the articles, which have been used in each topic.
- You have to be clear in the text, when you mention a gene or what it express, for example in line 58 you mention Pneumolysin together with lytA! lytA is a gene which express the Autolysin, while the ply gene express Pneumolysin! All mentioned gene names are normally written in italic!
- Line 310, S. Pneumoniae should be S. pneumoniae!
Author Response
Response to reviewer 2.
Many thanks for your continued advice.
- As you suggested I have added a Table 1. Data sources, which describes each data source.
- The gene names have been converted to italics.
S. pneumoniae has been thus corrected
Reviewer 3 Report
All comments and suggestions were addressed in the revised version of the manuscript.
Author Response
Thank you for your continued analysis of the article.
